# Repeatability of choriocapillaris flow voids by optical coherence tomography angiography in central serous chorioretinopathy

José Ignacio Fernández-Vigo[1,2]*, Francisco Javier Moreno-Morillo[1], Emilio López-Varela[3,4], Jorge Novo-Bujan[3,4], Marcos Ortega-Hortas[3,4], Bárbara Burgos-Blasco[1], Lorenzo López-Guajardo[1], Juan Donate-López[1]

1 Department of Ophthalmology, Hospital Clínico San Carlos, Instituto de Investigación Sanitaria (IdISSC), Madrid, Spain, 2 Department of Ophthalmology, Centro Internacional de Oftalmología Avanzada, Madrid, Spain, 3 Department of Computer Science, Centro de Investigacion CITIC, Universidade da Coruña, A Coruña, Spain, 4 Department of Computer Science, VARPA Research Group, Instituto de Investigación Biomédica de A Coruña (INIBIC), Universidade da Coruña, A Coruña, Spain

* jfvigo@hotmail.com

**Data Availability Statement:** All relevant data are within the paper.

## Abstract

### Purpose

To assess the repeatability of flow signal voids (FSV) measurements of the choriocapillaris (CC) and choroid (CH) in central serous chorioretinopathy (CSCR) by Swept-Source optical coherence tomography angiography (SS-OCTA).

### Methods

Cross-sectional study including 104 eyes of 52 patients with unilateral CSCR. Two consecutive macular 6x6 mm SS-OCTA scans (Plex Elite 9000; Zeiss, Dublin, CA) were obtained from the affected eyes with persistent subretinal fluid (SRF) (CSCR group) and the fellow unaffected eyes (control group). FSV area and the number of contours measurements were analyzed using three slabs: inner CC, outer CC and CH. The repeatability of the measurements was assessed with intraclass correlation coefficients (ICC) and coefficients of variation (CV).

### Results

In the CSCR group, ICCs for the FSV area in the three slabs were all ≥0.859, observing higher values for the outer CC and the CH (0.959 and 0.964) than for the inner CC (0.859). Similar ICC values were obtained for the FSV area in control eyes, observing the highest values for the outer CC (0.949), followed by the CH (0.932) and inner CC (0.844). Regarding the FSV number of contours measurements, ICCs were higher for the outer CC and CH (0.949 and 0.932) than for inner CC (0.844). CV for the FSV area was 4.7%, 3.8% and 8.6% in the CSCR eyes and 4.8%, 3.9% and 9.3% in the control group for the inner CC, outer CC and CH respectively.

**Funding:** The author(s) received no specific funding for this work.

**Competing interests:** The authors have declared that no competing interests exist.

## Conclusion

SS-OCTA offers good repeatability to quantify macular FSV in CSCR eyes and fellow eyes.

## Introduction

Central serous chorioretinopathy (CSCR) represents one of the main causes of visual acuity (VA) impairment in patients under 60 years. It is primarily characterized by subretinal fluid (SRF) associated with a retinal pigment epithelium (RPE) detachment in the posterior pole [1]. CSCR pathophysiology is not fully understood, although choroidal vessel dilation and choroidal hyperpermeability have been documented, thus including CSCR as part of the pachychoroid diseases [2–4]. In the latter there is a direct compression of the overlying choriocapillaris (CC) by the abnormally dilated choroidal vessels combined with limited elasticity or compliance [5, 6]. Choroidal imaging revealed thinning of the inner choroid and enlarged outer choroidal lumina in CSCR, further supporting this hypothesis. This may alter the barrier function of the CC–RPE complex, subsequently causing subretinal fluid (SRF) accumulation.

In this context, some authors have described an impaired flow of the CC and choroid in CSCR patients [7, 8]. Several authors have described that in CSCR irregular CC flow patterns corresponding to ICGA abnormalities are present [9–12]. Ho et al. reported that the flow deficit areas identified in the CC layer may suggest possible relative choroidal ischaemia [6].

The recently developed optical coherence tomography angiography (OCTA) uses different algorithms to detect red blood cell motions offering non-invasive visualization of the flow in different plexus of the retina and the choroid [13, 14]. The imaging and analysis of the CC using OCTA is challenging for multiple reasons. Firstly, the CC is a thin but dense vascular layer located in the inner choroid adjacent to the Bruch membrane (BM), and secondly, as Chu et al. stated, an accurate segmentation is crucial for correct visualization of the CC flow using OCTA [15]. Furthermore, the image has a grainy appearance of alternating white and black pixels being difficult to discern the capillary detail. This is thought to be due to the limited resolution of current OCTA technology, noise, as well as a dynamic aspect of the CC circulation [16].

Automatic measurement algorithms have been implemented in multiple OCT devices to obtain quantitative vessel measurements which enable us to describe the status or the changes of the retinal and choroidal vasculature [13, 17, 18]. These parameters include vessel density (VD), perfusion density or flow signal voids (FSV) [19]. As this technology is relatively new, an increasing interest in the reproducibility of OCTA measurements not only in the retina but also in the CC and choroid and in disease's eyes has been recently developed [20–23].

Therefore, the purpose of the present study is to assess the repeatability of SS-OCTA macular FSV measurements in CSCR eyes and fellow unaffected eyes.

## Methods

For this cross-sectional study, 52 patients with unilateral chronic CSCR were enrolled among those who came to a clinical visit over the period from January 10, 2020, to March 15, 2021, in Hospital Clínico San Carlos, Madrid (Spain). Both eyes of each patient were included to assess the reproducibility of the FSV analysis, divided into two groups: affected eyes with persistent subretinal fluid (SRF) for at least three months (CSCR group) and the fellow clinically unaffected eyes (control group).

Subjects were invited to participate if they met all the inclusion criteria and none of the exclusion criteria, after giving their written informed consent. The study protocol adhered to

the tenets of the Declaration of Helsinki and was approved by the Institutional Review Board of the Center.

The inclusion criteria were: healthy Caucasian subjects, aged ≥18 years, with a spherical refractive error between +3.0 and -3.0 diopters and intraocular pressure (IOP) <21 mm Hg in both eyes. Exclusion criteria were: ocular diseases (macular diseases such as age-related macular degeneration or macular dystrophies; retinal vascular diseases such as diabetic retinopathy, retinal vein occlusion or the presence of neovascular membrane; glaucoma and other neuropathies) or systemic pathology (such as arterial hypertension or diabetes), significant medium opacity of the lens or cornea, any ocular treatment in the three previous months or previous eye surgery. In addition, images with large motion artifacts due to the lack of collaboration and the impossibility to activate the eye tracker by the software were excluded.

The subjects enrolled first underwent a medical history and a comprehensive ophthalmologic examination followed by OCTA on the same day. The eye exam included visual acuity and refraction, slit-lamp biomicroscopy and posterior segment ophthalmoscopy. In each participant, age, gender and axial length (IOL master 700; Carl Zeiss Jena, Germany) were noted.

## OCTA examination

For the explorations the device employed was Plex Elite (Zeiss Meditec, Dublin, CA) a swept-source OCTA (SS-OCTA) which uses a central wavelength between 1,040 nm and 1,060 nm with an axial resolution of 6.3 μm, a transverse resolution of 20 μm and a scanning speed of 100,000 A-scans per second. The main characteristics of the 6x6 mm scan are: 500 A-scans, 500 B-scans, resolution of 12 μm, depth of 3 mm and pixel depth of 1536 pixels. The algorithm employed was AngioPlex Elite 9000 (AngioPlex Elite 9000, Zeiss, Germany). All OCTA images were acquired by two well-trained examiners (FJMM and JIFV). Only images of sufficient quality, as determined by a signal quality >7/10, were accepted.

Two consecutive macular scans centered in the macula with an area of 6 x 6 mm separated by a 2 minutes interval using the SS-OCTA device were obtained. The eye tracker was activated, so the exploration was automatically centered on the same area. Therefore, the two scans were aligned before the analysis. Three slabs were analyzed: inner choriocapillaris (CC) (extends 4 to 20 μm below the retinal pigment epithelium, RPE), outer CC (extends 29 to 49 μm below the RPE) and choroid (CH; measured from 64 to 115 μm below the RPE) (Fig 1).

These slabs (outer CC and choroid) were defined based on the default settings of the software device, and the inner CC was based on the proposal by different expert authors [15]. FSV area and the number of contours measurements were analyzed using customized software (Figs 2 and 3).

Images were analyzed after checking for appropriate segmentation and applying the device's software projections removal algorithm.

## Computational image analysis

All the images were processed using a custom automatic algorithm developed in Python 3.7.9 to detect and quantify the presence of FSV in the CC slab and CH slab [3, 12, 24, 25]. In addition to Python, the open-source library OpenCV 4.5.1.48 was used for the development of this algorithm [26]. A diagram showing the different steps of this image-processing algorithm is presented in Fig 4.

For each of our images, we extracted a binary mask where the different areas corresponding to the FSV were represented. To extract this binary mask, first, the images were normalized to a common range of values using a min-max normalization. The obtained values were mapped to the common range (0–255) using linear scaling. This normalization of the range of values

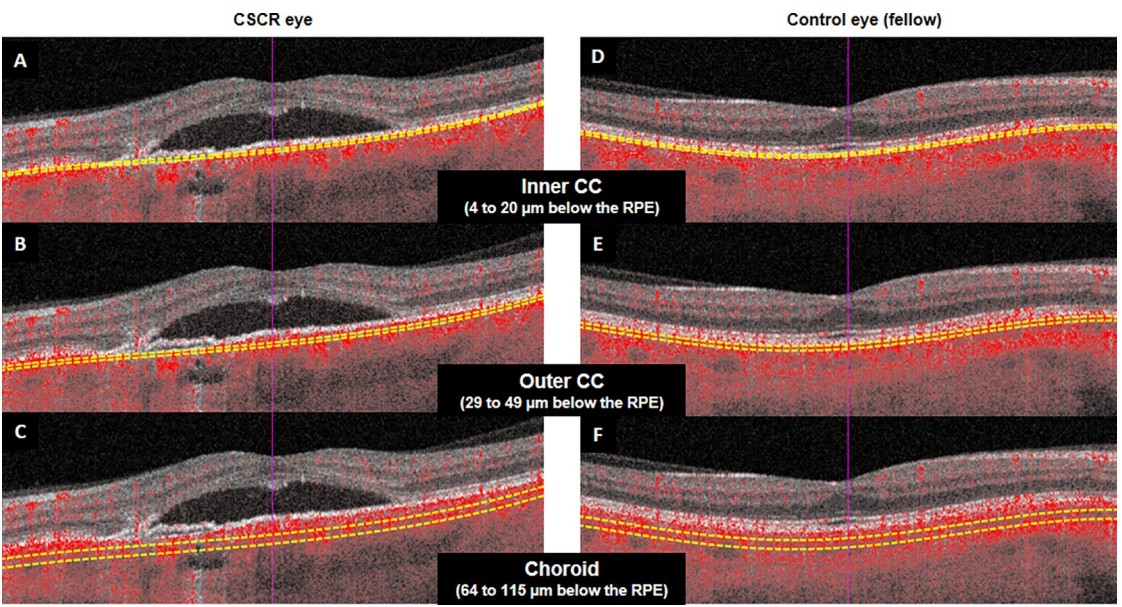

**Fig 1.** This image shows the B-scan of the optical coherence tomography with flow superposition and the three different slabs studied segmented: 1) inner choriocapillaris (CC, 4 to 20 µm below the retinal pigment epithelium, RPE), 2) outer CC (29 to 49 µm below the RPE) and 3) choroid (CH, 64 to 115 µm below the RPE) in central serous chorioretinopathy (CSCR) and fellow eyes.

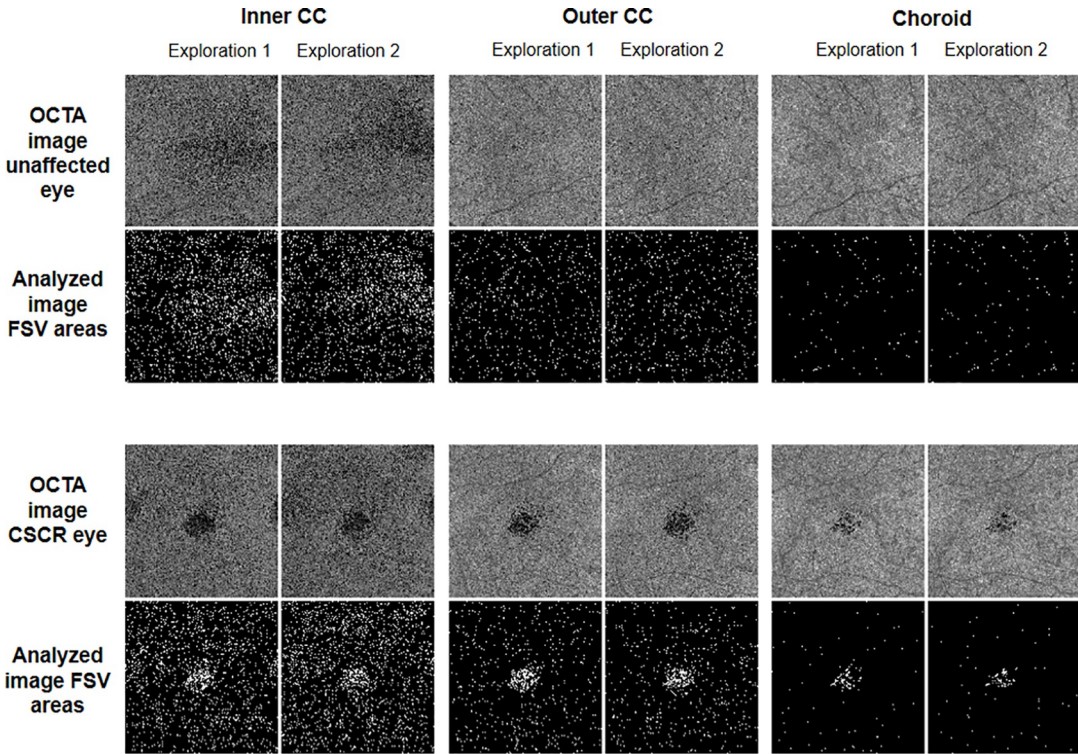

**Fig 2. Optical coherence tomography angiography (OCTA) images and the corresponding analyzed images of the flow signal voids (FSV) area of the three slabs studied (inner choriocapillaris, CC; outer CC and choroid) for the exploration 1 and 2 (two consecutive scans separated by a two-minute interval) in central serous chorioretinopathy (CSCR) and fellow eyes.**

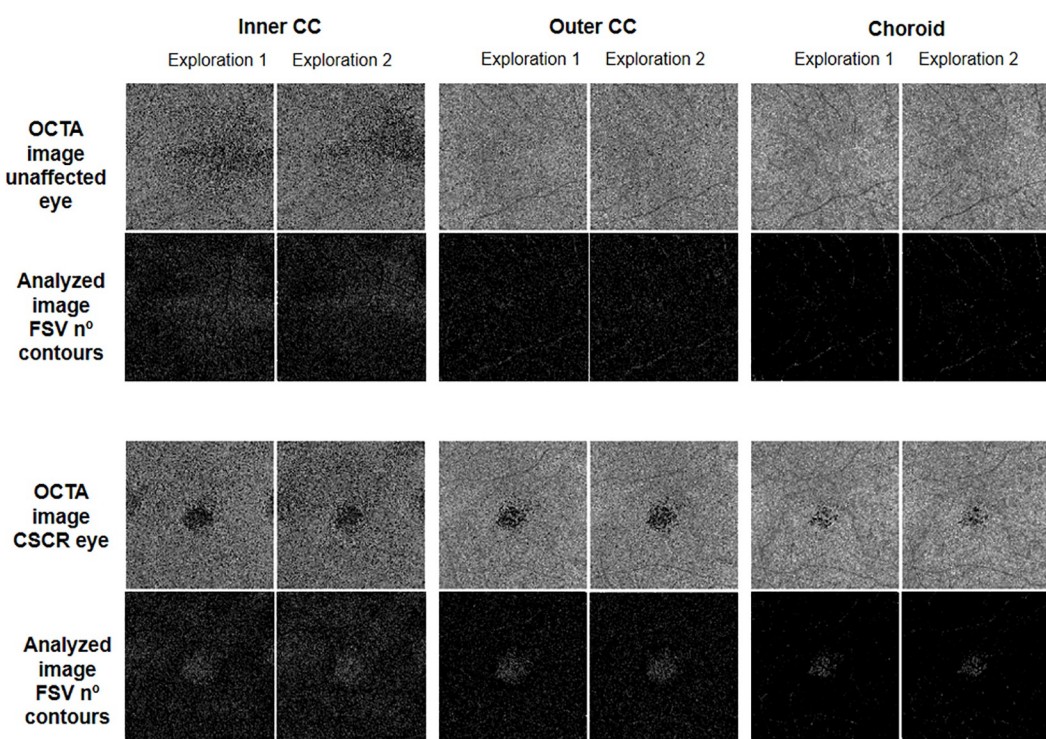

**Fig 3. Optical coherence tomography angiography (OCTA) images and the corresponding analyzed images of the flow signal voids (FSV) number of contours of the three slabs studied (inner choriocapillaris, CC; outer CC and choroid) for the explorations 1 and 2 in central serous chorioretinopathy (CSCR) and fellow eyes.**

provides robustness to the image processing, which increases the repeatability of the method. Then, a contrast-limited adaptive histogram equalization (CLAHE) with a neighborhood size (small tiles) of 8x8 was applied. The application of CLAHE allows us to increase the contrast between the darkest and brightest areas of the image locally. This mitigates the problems caused by illumination changes in small areas of the image and allows us to segment the FSV more accurately. Next, inverse global thresholding was applied, resulting in a binary image where the flow voids were marked in white. To establish the threshold value, the effect of different values was tested to maximize the flow voids to noise ratio. The higher the threshold value, the higher the probability of false positives, so the smallest value that allows segmenting the true flow voids accurately was selected as the threshold value. This value was set to 30, as it produces an appropriate balance between flow voids segmentation and noise minimization. Lastly, a filter of the detected flow voids was performed on this binary image, keeping those with an area greater than 20 pixels (0,686 µm2) after the application of a soft dilation. This filter size allows us to remove the noise that is present in the binary image and helps us keep only those FSV that are significant. To select the size of this filter size we used the same criteria as with the threshold value. Therefore, we used the largest filter size that allowed us to eliminate false positives without degrading the true flow voids. Finally, from this binary mask we calculated two different parameters. On the one hand, we calculated the total area of FSV. This value corresponds to the sum of all relative areas that do not have blood flow in the image. On the other hand, we calculated the number of FSV contours corresponding to the number of flow voids regions present in the image. These values were transformed from pixels to $\mu m^2$.

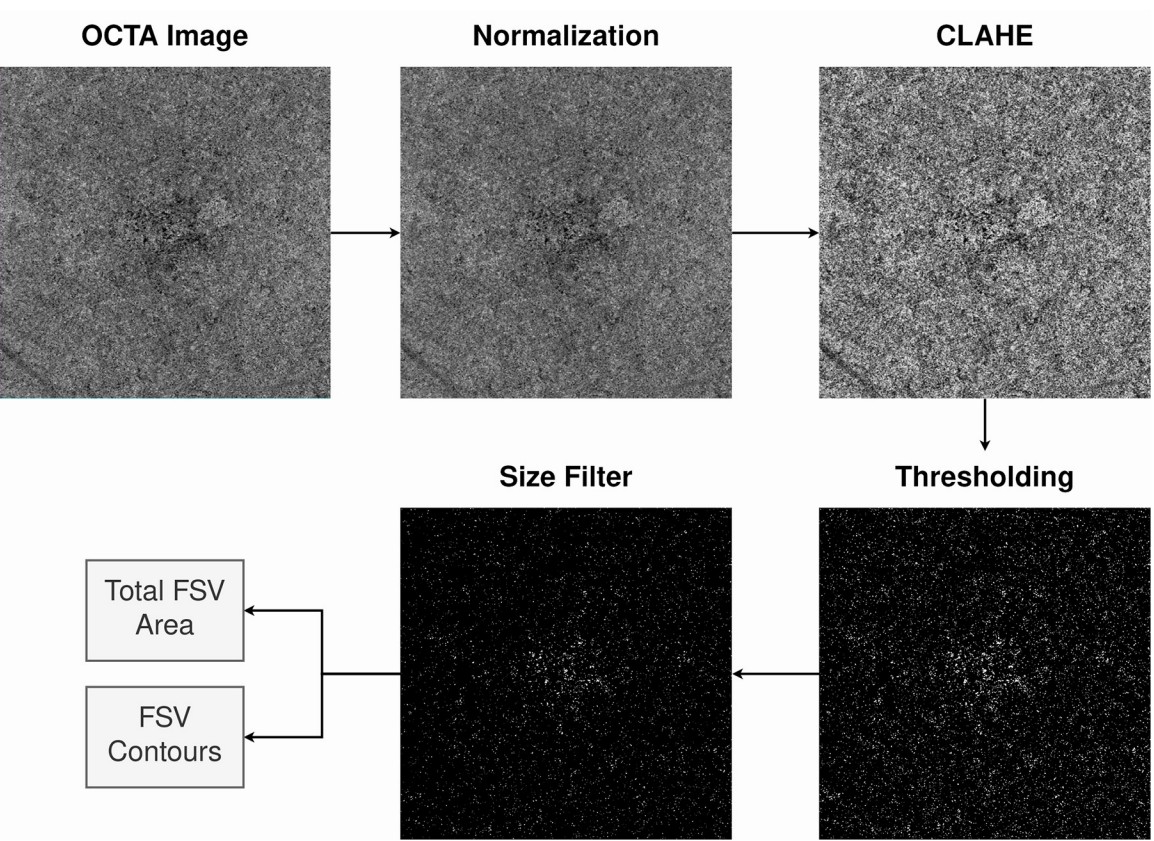

**Fig 4. Diagram showing the different steps of this image-processing algorithm.**

## Statistical analysis

All statistical tests were performed using the software package SPSS® (Statistical Package for Social Sciences, v21.0; SPSS Inc., Chicago, IL, USA). Quantitative data are provided as the mean and standard deviation. Qualitative data are expressed as their frequency distributions. In the FSV repeatability analysis, for each measurement, the intraclass correlation coefficient (ICC; two-way mixed effects, absolute agreement, single measurement and mean measurement) with its confidence interval 95% (IC95) was calculated for the two consecutive scans. The coefficient of variation (CV), a statistical measure of the relative dispersion of data points in a data series around the mean, was also calculated. Lastly, Bland-Altman plots were calculated to analyze the agreement between the measurements in the FSV area and number of contours in the different segmentations studied. Significance was set at $p < 0.05$.

## Results

Mean participant age was 48.5 ±8.7 years (range, 22–64 years); 60.5% were males. Mean AL was 23.3 ±1.1 mm (range, 20.1–25.5 mm). Two cases were excluded due to a lack of collaboration which resulted in motion artifacts, being the final sample size analyzed one hundred and four eyes of 52 patients with unilateral CSCR, including 52 affected eyes (with the presence of SRF) and 52 eyes clinically unaffected.

The mean FSV area in the CSCR group was 5.319 ± 0.934 $\mu m^2$ (range 3.041–7.645) and 5.316 ± 0.942 (3.473–8.008) for the inner CC, 1.998 ± 0.557 (1.136–3.899) and 1.998 ±

**Table 1. Absolute values and repeatability of flow signal voids (FSV) area (μm$^2$) in the different groups and slabs studied by swept-source optical coherence tomography angiography.**

| CSCR eyes (n = 52 eyes) | Inner CC | Outer CC | Choroid |
|---|---|---|---|
| Measurement 1 | 5.319 ± 0.934 | 1.998 ± 0.557 | 0.538 ± 0.349 |
| | (3.041–7.645) | (1.136–3.899) | (0.193–2.056) |
| Measurement 2 | 5.316 ± 0.942 | 1.998 ± 0.576 | 0.531 ± 0.333 |
| | (3.473–8.008) | (1.095–4.018) | (0.157–1.850) |
| ICC single measures | 0.859 | 0.959 | 0.964 |
| | (0.763–0.918) | (0.928–0.977) | (0.937–0.979) |
| ICC average measures | 0.924 | 0.979 | 0.982 |
| | (0.866–0.957) | (0.963–0.988) | (0.967–0.990) |
| CV | 4.71% | 3.82% | 8.62% |
| Fellow eyes (n = 52 eyes) | Inner CC | Outer CC | Choroid |
| Measurement 1 | 5.279 ± 0.966 | 1.736 ± 0.400 | 0.438 ± 0.315 |
| | (3.517–9.776) | (1.102–3.637) | (0.159–2.306) |
| Measurement 2 | 5.249 ± 0.869 | 1.729 ± 0.399 | 0.465 ± 0.350 |
| | (3.334–8.226) | (1.071–3.430) | (0.110–2.627) |
| ICC single measures | 0.844 | 0.949 | 0.932 |
| | (0.744–0.908) | (0.913–0.970) | (0.885–0.961) |
| ICC average measures | 0.916 | 0.974 | 0.965 |
| | (0.853–0.952) | (0.954–0.985) | (0.939–0.980) |
| CV | 4.78% | 3.86% | 9.30% |

CSCR = central serous chorioretinopathy; CC = choriocapillaris; ICC = intraclass correlation coefficient (95% confidence interval); CV = coefficient of variation.

0.576 μm$^2$ (1.095–4.018) for the outer CC and 0.538 ± 0.349 (0.193–2.056) and 0.531 ± 0.333 μm$^2$ (0.157–1.850) for the CH for the measurement 1 and 2 respectively (Table 1).

The ICC was highest for the choroid 0.964 (IC95, 0.937–0.979), followed by the outer CC 0.959 (0.928–0.977), and being the lowest the inner CC 0.859 (0.763–0.918).

In the fellow eyes (control group) the mean FSV area for the inner CC was 5.279 ± 0.966 (range 3.517–9.776) and 5.249 ± 0.869 (3.334–8.226); for the outer CC 1.736 ± 0.400 (1.102–3.637) and 1.729 ± 0.399 (1.071–3.430); and for the CH 0.438 ± 0.315 (0.159–2.306) and 0.465 ± 0.350 (0.110–2.627) (Table 1). The highest ICC was observed for the outer CC 0.949 (IC95, 0.913–0.970), followed by the CH 0.932 (0.885–0.961) and being the lowest ICC for the inner CC 0.844 (0.744–0.908).

For the FSV number of contours in the CSCR eyes the highest ICC was observed for the CH 0.935 (0.887–0.963) followed by the outer CC 0.884 (0.802–0.934) and by the inner CC 0.832 (0.718–0.902) (Table 2).

Similar ICCs values were observed between CSCR eyes and the fellow eyes (0.859, 0.959 and 0.964 versus 0.844, 0.949 and 0.932 for the inner CC, outer CC and choroid respectively).

In the fellow eyes, higher ICC values were observed for the outer CC 0.907, followed by the CH 0.881, and by the inner CC 0.843.

In the comparison between CSCR eyes and the fellow eyes for the inner and outer CC measurements, similar ICCs values were observed (0.832 and 0.884 versus 0.843 and 0.907 respectively). However, a noticeably higher ICC value was observed for the disease eyes than for the unaffected ones (0.935 vs 0.881 respectively).

Regarding the laterality, 54 right and 50 left eyes were affected by CSCR, and no differences were observed in terms of the ICCs values.

**Table 2. Absolute values and repeatability of flow signal voids (FSV) number of contours in the different groups and slabs studied by swept-source optical coherence tomography angiography.**

| CSCR eyes (n = 52 eyes) | Inner CC | Outer CC | Choroid |
|---|---|---|---|
| Measurement 1 | 13,975 ± 1,104 (11,332–16,568) | 11,055 ± 986 (8,843–14,180) | 4,115 ± 1,487 (1,636–10,328) |
| Measurement 2 | 14,093 ± 1,188 (11,005–16,476) | 11,038 ± 990 (8,600–14,123) | 4,077 ± 1,491 (1,544–10,201) |
| ICC single measures | 0.832 (0.718–0.902) | 0.884 (0.802–0.934) | 0.935 (0.887–0.963) |
| ICC average measures | 0.908 (0.836–0.949) | 0.939 (0.890–0.966) | 0.966 (0.940–0.981) |
| CV | 2.24% | 1.97% | 6.46% |
| **Fellow eyes** (n = 52 eyes) | Inner CC | Outer CC | Choroid |
| Measurement 1 | 13,975 ± 1,179 (9,039–16,301) | 11,088 ± 972 (9,144–14,731) | 3,925 ± 1,513 (1,858–10,718) |
| Measurement 2 | 14,117 ± 998 (3,334–8,226) | 11,097 ± 1,064 (8,933–14,777) | 4,051 ± 1,589 (1,247–11,140) |
| ICC single measures | 0.843 (0.742–0.907) | 0.907 (0.843–0.946) | 0.881 (0.801–0.930) |
| ICC average measures | 0.915 (0.852–0.951) | 0.951 (0.915–0.972) | 0.937 (0.890–0.964) |
| CV | 2.25% | 1.92% | 8.91% |

CSCR = central serous chorioretinopathy; CC = choriocapillaris; ICC = intraclass correlation coefficient (95% confidence interval); CV = coefficient of variation.

The CV for the FSV area was 4.71%, 3.82% and 8.62% in the CSCR group and 4.78%, 3.86% and 9.30% in the non-CSCR group for the inner CC, outer CC and CH respectively (Table 1). For the FSV number of contours, the CV was 2.24%, 1.97% and 6.46% in the CSCR group and 2.25%, 1.92% and 8.91% in the control group for the inner CC, outer CC and CH respectively (Table 2).

Bland-Altman plots for the FSV area (Fig 5) and number of contours (Fig 6) showed a good agreement between the measurements performed in the different segmentations studied.

## Discussion

The recent combination of quantitative analysis and OCTA technology implies that the reproducibility of the measurements, including FSV, must be determined so that this technique can

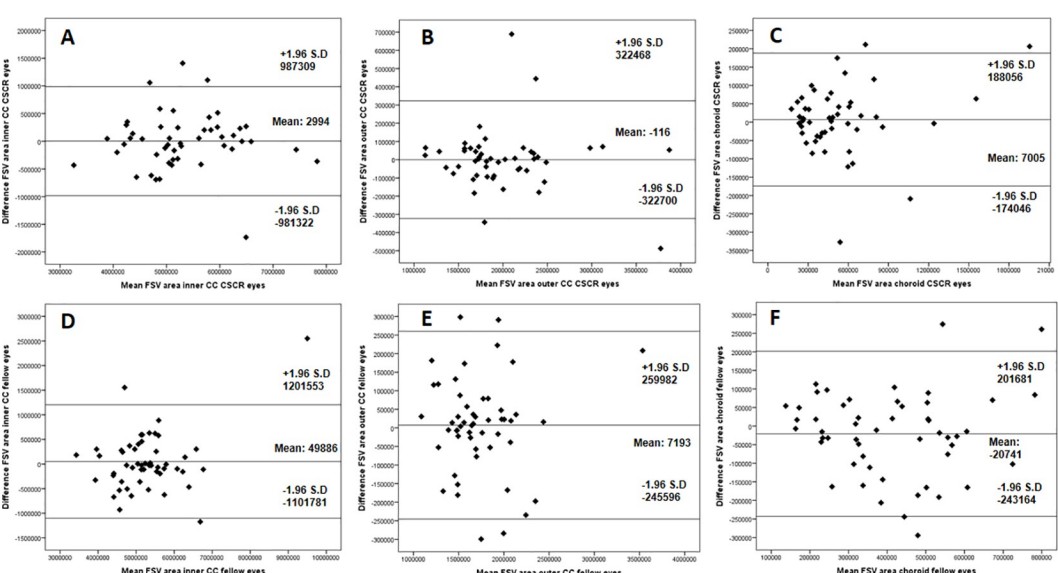

**Fig 5.** Bland-Altman plots showing the flow signal voids (FSV) area in the different segmentations studied (A and D: inner choriocapillaris (CC) in CSCR and fellow eyes respectively; B and E: outer CC in CSCR and fellow eyes respectively, C and F: choroid in CSCR and fellow eyes respectively).

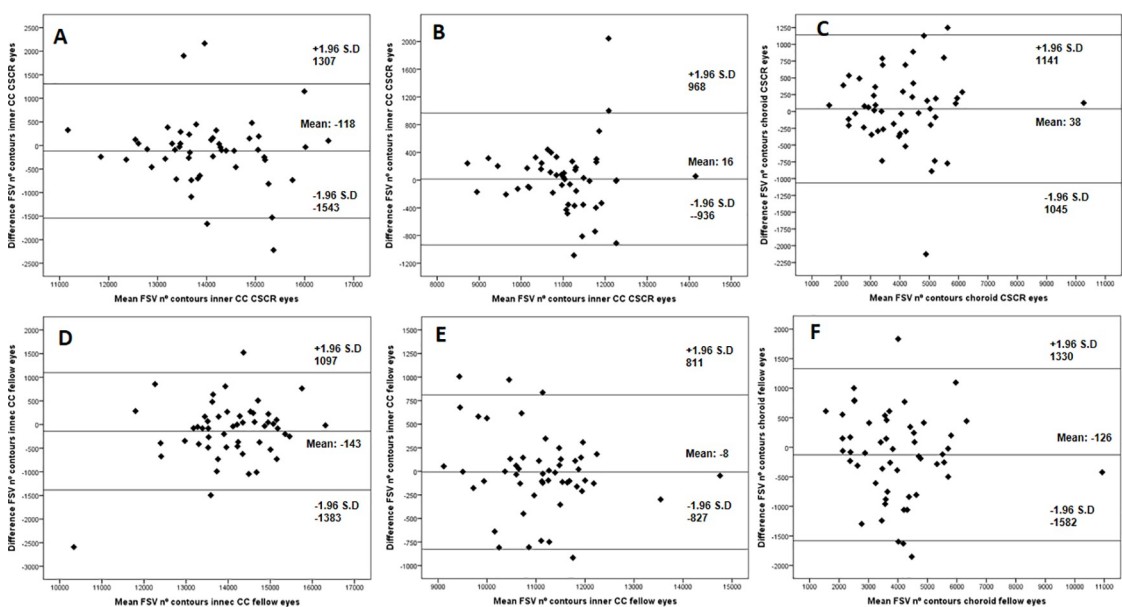

**Fig 6.** Bland-Altman plots showing the flow signal voids (FSV) number of contours in the different segmentations studied (A and D: inner choriocapillaris (CC) in CSCR and fellow eyes respectively; B and E: outer CC in CSCR and fellow eyes respectively, C and F: choroid in CSCR and fellow eyes respectively).

be implemented in clinical research and practice [8, 25, 27]. Moreover, in order to use OCTA CC flow metrics as a biomarker for assessing disease and progression, it should be able to demonstrate good repeatability in both normal and pathological eyes.

In this study, we used an SS-OCTA device and developed a new algorithm to quantify in two different manners: FSV area and the number of contours in the CC and choroid in CSCR patients, both in clinically affected and unaffected eyes. Our data indicate good repeatability for all the slabs (ICC ≥0.844), being higher in the CH and the outer CC than in the inner CC. Similar ICC values were observed between CSCR eyes and unaffected fellow eyes, except for the choroid in which higher values were observed for the disease's eyes.

It is well-known that the CC is a very thin vascular layer and the CC images currently provided by OCTA show a granular pattern of bright and dark areas of different sizes, individual capillaries in the CC not being able to be identified [5, 14]. Brighter regions represent higher flow areas while dark areas are called flow voids and depict areas where there is a lack of flow signal. Besides showing comparable lateral resolution to structural OCT, OCTA can detect CC blood flow, producing contrast between the RPE and CC [13, 14]. Lin et al. have recently described that, as opposed to subfoveal choroidal thickness, there does not appear to be significant diurnal variation in CC flow voids, in terms of the density, size and numbers, in normal individuals. In their study, the CC image was taken from 31 μm to 40 μm below the RPE. Thus, the aforementioned study suggests that alterations of CC flow deficit seen in pathological eyes will not be confounded by the diurnal fluctuation [28].

We have studied two different CC slabs to observe which one offers the most repeatable results. The first one is offered by default by the device's software (29 to 49 μm below the RPE). More recently, a new inner slab (4 to 20 μm below the RPE) has been proposed by different authors given that the true anatomical location of the CC lies immediately under the RPE and BM [15]. In our study, we have observed higher repeatability values for the outer CC than the inner CC. This could be due to the well-known RPE signal scattering and an artifact image of

the CC when the slab is segmented immediately below the RPE. The choroid slab offered good repeatability both in CSCR and fellow eyes (0.964 and 0.932) being probably less dependent on projection artifacts. However, it presents the higher CV, which means a greater level of dispersion around the mean in the CH measurements.

Byon et al. have studied 12 healthy subjects employing a 3x3-mm scan using the same device with three 10-mm-thick slabs starting 11, 21, and 31 mm below the RPE, observing an excellent ICC of 0.963, 0.975, and 0.911 respectively. They concluded that regardless of which parameter was modulated, the 21-31-μm slab was the most repeatable. Interestingly, in accordance with our study, Byon et al. have described that in some cases the most inner slab (11-21-μm) demonstrated a hypointense region caused by inadvertent inclusion of the RPE, so they decided to not included this slab in subsequent analyses [16].

In a previous study carried out by our group using another SS-OCTA device (Triton, Topcon, Nagoya, Japan) for the reproducibility of the vessel density quantification in healthy individuals, the CC at the macula, segmented from RPE to 20.2 μm beneath it similar to the inner CC studied here, showed good repeatability in the foveal subfield (ICC = 0.718) [23]. However, poor repeatability was found in the parafoveal sectors (≤0.499). It should be highlighted that Yun et al. described that VD and FV areas of the CC varied according to the device used and the image adjustment method [29].

Regarding CSCR, Xu et al. have described an aberrant flow in this disease, with a flow pattern described as focally increased and decreased pixel values, which implies a coexisting increased and decreased flow in the CC [11]. However, the main limitation of that study, as the authors recognized, was a qualitative evaluation due to the inability to get a quantitative analysis of the aberrant flow, and therefore, the lack of reproducibility of this assessment. Rochepeau et al. have demonstrated CC flow reduction in the unaffected eyes of patients with acute, recurrent, or persistent CSC at onset compared to age-matched healthy individuals, suggestive of a primary choroidopathy including ischemic processes [8].

Yang et al. studied 56 CSCR eyes, observing a significant decrease in the total FSV area at 6 months after photodynamic therapy [25]. They have also studied an interesting issue about the possible influence of the SRF in the attenuation of FSV. They described that the relationship between the mean total area of FSV and SFCT, subfoveal CC layer thickness, and subfoveal choroidal large vessel layer thickness was not largely influenced by the presence of SRF. In the present study, we have studied two different groups, the CSCR eyes with SRF and the clinically unaffected eyes used as controls to assess the possible influence of the SRF in the FSV quantification at the CC and CH. We have observed very similar repeatability in eyes with and without SRF, so it seems that the FSV measurement is indeed a reliable parameter. Burnasheva et al. described that the presence of the general decrease of CC perfusion in both eyes of CSCR patients was irrespective of the presence of SRF or asymptomatic structural RPE changes [30]. Remarkably, as described by Byon et al. in diseased eyes, the RPE fit would generally be preferred for CC assessment, as it would essentially flatten the reference line [16].

Therefore, as Spaide also described [27], a reproducible quantitative analysis of the choriocapillaris was generally possible using FSV in CSCR patients with or without SRF. This is also interesting because different authors such as Teussink et al. described that the abnormal vasculature in the CC layer persisted even after the resolution of SRF [9]. In this regard, Reich et al. have described that the presence of SRF could be an important shadow-causing artifact source for CC OCTA analysis which can be mitigated but not completely eliminated by employing SS-OCTA, being the latter employed in the present study [31].

This is the first study assessing the repeatability of the FSV quantification in the CC and choroid in CSCR patients, both in clinically affected and unaffected eyes. The clinical relevance of these findings is that this instrument could be useful to reliably assess changes in the

perfusion of the CC and choroid during follow-up or after different treatment approaches in clinical research and practice. For example, a recent study with high interest was carried out by Ho et al. comparing PDT versus micropulse laser (MLT) in CSCR patients [6], observing that PDT has a stronger effect than MLT in promoting CC flow deficit area recovery. Also, the safety and side effects of different treatments applied over the CC and choroid such as the PDT and its recanalization [32] could be reliably assessed given that the analysis of the FSV has now been found to be reproducible.

Our study has several limitations, mainly inherent to the OCTA technique. Firstly, quantification of the CC employing an OCTA device is limited by the projection artefact of the retinal and the RPE scattering, which may affect the accuracy of the measurements, even with the device's projection removal software [17]. Secondly, multiple factors including the methods used, motion artifacts, image quality or the device threshold for the detection of flow may alter the accuracy and repeatability of OCTA measurements such as FSV [20, 33, 34]. In addition, while most authors agree that measurements taken with the same OCTA machine and technique are reproducible, there is significant variability between different devices and techniques [16, 29]. Also, the optimal anatomical slab position for the CC assessment should be theoretically segmented just below the BM and extend to the inner border of the Sattler layer. However, this correct segmentation is currently still challenging [15, 16]. An hypothetical alternative for the analysis of the CC could be the measurements from Sattler's layer based on histology. However, defining the outer choroidal vessel layers is not always clearly defined. In future studies, refining the algorithms employed by these devices will improve the reproducibility of this technology for the quantification of FSV.

In conclusion, SS-OCTA offers good repeatability to quantify macular FSV, both in CSCR eyes, even with the presence of SRF, and fellow eyes. Both segmentation slabs for the CC showed good reproducibility, being higher for the outer CC. Choroidal FSV has also good repeatability but a higher CV. Therefore, this instrument could be useful to reliably assess changes in the perfusion of the CC and choroid during follow-up or after different treatment approaches.

## Author Contributions

**Conceptualization:** José Ignacio Fernández-Vigo, Jorge Novo-Bujan, Marcos Ortega-Hortas, Lorenzo López-Guajardo, Juan Donate-López.

**Data curation:** José Ignacio Fernández-Vigo, Francisco Javier Moreno-Morillo, Emilio López-Varela, Jorge Novo-Bujan, Marcos Ortega-Hortas, Lorenzo López-Guajardo.

**Formal analysis:** José Ignacio Fernández-Vigo, Francisco Javier Moreno-Morillo, Emilio López-Varela, Jorge Novo-Bujan, Marcos Ortega-Hortas, Bárbara Burgos-Blasco, Lorenzo López-Guajardo, Juan Donate-López.

**Investigation:** José Ignacio Fernández-Vigo, Emilio López-Varela, Bárbara Burgos-Blasco, Juan Donate-López.

**Methodology:** José Ignacio Fernández-Vigo, Francisco Javier Moreno-Morillo, Emilio López-Varela, Jorge Novo-Bujan, Marcos Ortega-Hortas, Bárbara Burgos-Blasco.

**Software:** José Ignacio Fernández-Vigo, Emilio López-Varela, Jorge Novo-Bujan, Marcos Ortega-Hortas.

**Supervision:** José Ignacio Fernández-Vigo, Francisco Javier Moreno-Morillo, Emilio López-Varela, Marcos Ortega-Hortas, Bárbara Burgos-Blasco, Lorenzo López-Guajardo, Juan Donate-López.

**Validation:** José Ignacio Fernández-Vigo.

**Writing – original draft:** José Ignacio Fernández-Vigo, Francisco Javier Moreno-Morillo, Bárbara Burgos-Blasco, Lorenzo López-Guajardo, Juan Donate-López.

**Writing – review & editing:** José Ignacio Fernández-Vigo, Francisco Javier Moreno-Morillo, Emilio López-Varela, Jorge Novo-Buján, Marcos Ortega-Hortas, Bárbara Burgos-Blasco, Lorenzo López-Guajardo, Juan Donate-López.

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
