## [Decision Letter · Decision Letter 0]

20 Oct 2022

PONE-D-21-35561Repeatability of choriocapillaris flow voids by optical coherence tomography angiography in central serous chorioretinopathy.PLOS ONE

Dear Dr. Fernández-Vigo,

Thank you for submitting your manuscript to PLOS ONE. After careful consideration, we feel that it has merit but does not fully meet PLOS ONE’s publication criteria as it currently stands. Therefore, we invite you to submit a revised version of the manuscript that addresses all points raised during the review process. I also sincerely apologise for the very long delay in the review process.

A main limitation in the study is that full details of the methods used and the criteria for analyses are not fully presented or explained. These details are important to establish whether the method is repeatable and can be applied by other investigators.

Please see below comments that should be addressed; there are both major and minor issues.

Major Comment

1. Page 6, line 117 “All OCTA images were acquired by two well-trained examiners (FJMM and JIFV). Only images of sufficient quality, as determined by a signal quality >7/10, were accepted.” Please provide details on intra- and inter-observer image acquisition. Is the >7/10 signal quality used here, specific to the OCTA instrument used in this study, or a general feature? The criteria for a quality image are relevant in establishing a useful method. Thank you.

2. Page 8, line 148 “FSV was defined as the proportion of absence of blood flow relative to the total area measured.” Please clarify that FSV used in the analyses was not individual areas but rather a relative area that did not have blood flow per total macula area measured.

3. Page 8, line 158 “Different combinations of thresholds were tested and a threshold of 30 was selected as the most appropriate. Last, a filter of the detected 160 flow voids was performed on this binary image, keeping those with an area greater than 20 pixels (0,686 μm2) after the application of a soft dilation.”

Line 163 “Different area sizes were tested when applying the size filter and the most appropriate one was selected.”

For the above parameters, how was ‘appropriate’ defined? Why was 20pixels selected as the cut-ff for the FSV area?

4. A summary flow diagram indicating the series of processes used, the two algorithms applied and decisions to establish the final method should be included. These further details for defining the method used.

5. Page 15, Discussion. The comments on the effects of subretinal fluid in CSCR (and presumably other conditions) are very relevant and useful for clinical applications of OCTA. It would be useful to assess the same eyes with CSCR after resolution of the subretinal fluid (not just compare with unaffected other eyes of each patient). Although this is likely not practical, can the authors comment on any observations they have made post-resolution of SRF in their patients in relation to their technique and repeatability. Thank you.

6. The three levels of image slabs taken all start from the RPE and are measured vertically into the underlying choriocapillaris and choroid. May have missed this, but how many scans are included for each slab? The two choriocapillaris slabs are 15 micron and 20 microns respectively, measured from the RPE/Bruch’s membrane location. The authors do mention whether taking measurements from Satller’s layer may be helpful however, based on histology, defining the outer choroidal vessel layers is not always clearly defined.

Minor comments

1. Please further clarify the exclusion criteria applied for selecting participants in the study, including details on types of pathology or disease. Were differences between right and left eyes, and gender analysed?

2. Line 150: “.. range of values using a mix max..” – should this be “min-max” here?

3. Line 153: What are the units for the 8x8 mentioned here?

We look forward to receiving your revised manuscript.

Kind regards,

Michele Madigan

Academic Editor

PLOS ONE

Journal Requirements:

1, Please ensure that your manuscript meets PLOS ONE's style requirements, including those for file naming. The PLOS ONE style templates can be found at

Reviewers' comments:

Reviewer's Responses to Questions

**Comments to the Author**

1. Is the manuscript technically sound, and do the data support the conclusions?

Reviewer #1: Yes

2. Has the statistical analysis been performed appropriately and rigorously? 

Reviewer #1: Yes

3. Have the authors made all data underlying the findings in their manuscript fully available?

Reviewer #1: No

4. Is the manuscript presented in an intelligible fashion and written in standard English?

Reviewer #1: Yes

5. Review Comments to the Author

Reviewer #1: Manuscript Number PONE-D-21-35561

“Repeatability of choriocapillaris flow voids by optical coherence tomography angiography in central serous chorioretinopathy"

This is a cross-sectional study which examined the intra-visit repeatability of the choroidal flow voids eyes with CSR. The study is an imaging methodological paper on choriocapillaris images in CSR eyes.

Major comments

1. Please provide the inter-grader (2 persons extracting the same slab from the same scan) and intra-grader (1 person extracting the same slab from the same scan twice) reliability.

2. Please provide the definition or the standard of appropriate: “Different combinations of thresholds were tested and a threshold of 30 was selected as the most appropriate. Different area sizes were tested when applying the size filter and the most appropriate one was selected.”

3. Please provide a rationale for analysis of the three slabs at this specific layers/thickness/depth.

4. Please define FSV area – is this the size of single flow void or the total area of flow void in the scan. The authors have only showed the FSV area. Please also show the density of flow void.

5. Please compare the results between normal and CSR eyes.

6. Please also show the Bland–Altman plots of the different choriocapillaris flow void metrics.

Minor comments

7. Please provide a step-by-step flow chart on how the image processing was carried out.

8. Were the two scans aligned before the analysis? Previous study (PMID: 31833241) used a customized software to align the scan prior to analysis. Was the eye tracker turned on during the scanning of the two consecutive macular scans?

9. Please elaborate the types of pathologies excluded from the study. “Exclusion criteria were ocular or systemic pathology…”

10. Were scans with other types of artifacts i.e., motion or poor segmentation excluded from the study? “Only images of sufficient quality, as determined by a signal quality >7/10, were accepted.”

11. Figure 2. Please clarify exploration 1 and 2.

6. PLOS authors have the option to publish the peer review history of their article (what does this mean?). If published, this will include your full peer review and any attached files.

Reviewer #1: No

---

## [Author Response · Author response to Decision Letter 0]

12 Nov 2022

PONE-D-21-35561

Repeatability of choriocapillaris flow voids by optical coherence tomography angiography in central serous chorioretinopathy.

PLOS ONE

Dear Dr. Fernández-Vigo,

Thank you for submitting your manuscript to PLOS ONE. After careful consideration, we feel that it has merit but does not fully meet PLOS ONE’s publication criteria as it currently stands. Therefore, we invite you to submit a revised version of the manuscript that addresses all points raised during the review process. I also sincerely apologise for the very long delay in the review process.

A main limitation in the study is that full details of the methods used and the criteria for analyses are not fully presented or explained. These details are important to establish whether the method is repeatable and can be applied by other investigators.

Dear Editor, 

First, we would like to thank you and the reviewer for this thorough revision of our manuscript. We feel that the comments made and the changes suggested have certainly helped improve the description of our work. All changes made in the manuscript have been highlighted in blue. We have improved the description of the methods employed.

Each of the coauthors has seen and agrees with each of the changes made to this manuscript in the revision. Please do not hesitate to contact me if you require any further information about our work.

Sincerely yours,

José Ignacio Fernández-Vigo

Please see below comments that should be addressed; there are both major and minor issues.

Major Comment

1. Page 6, line 117 “All OCTA images were acquired by two well-trained examiners (FJMM and JIFV). Only images of sufficient quality, as determined by a signal quality >7/10, were accepted.” Please provide details on intra- and inter-observer image acquisition. Is the >7/10 signal quality used here, specific to the OCTA instrument used in this study, or a general feature? The criteria for a quality image are relevant in establishing a useful method. Thank you.

Response: Thanks for the assessment of the manuscript performed and for the opportunity to clarify different issues.

Each of the patients included in the present study was examined by only one examiner (FJMM or JIFV). The process to acquire images is really automatic and examiner independent. The patient has to put the chin and forehead in the OCT device, after that the examiner selects the exploration mode (in this case 6x6 mm OCTA). Later, the explorer selects the button autofocus and optimize and the device center automatically the scan and focuses on the macular area. Finally, the explorer clicks on the start button for scanning the macula when the signal quality is green (>7/10). Therefore, there is no intra or inter-grader assessment of the images acquired, because is highly an automatic exploration.

As the reviewer remarks, signal quality is a relevant factor in OCTA studies as many authors have shown. Very recently, Dastiridou et al. have remarked that the SSI is a significant variable that explains the variation in the retina microvasculature in the macula and the optic disc, and, in the CC, a larger SSI was associated with a lower vessel density. Their findings underscore the importance of controlling for SSI in studies using OCTA.

Each OCTA instrument has its own signal quality indicator. In the device employed here (PlexElite, Zeiss Meditec, Dublin, CA), this should be >7/10. For example, in the DRI-Triton® (Topcon Corporation, Tokyo, Japan) images considered of good quality should have a signal strength intensity (SSI) above 40. For the Spectralis device (Heidelberg Engineering, Heidelberg, Germany), images should have an SSI above 25 (in a range from 0 to 45).

Dastiridou A, et al. Age and signal strength-related changes in vessel density in the choroid and the retina: an OCT angiography study of the macula and optic disc. Acta Ophthalmol. 2022;100(5):e1095-e1102. 

2. Page 8, line 148 “FSV was defined as the proportion of absence of blood flow relative to the total area measured.” Please clarify that FSV used in the analyses was not individual areas but rather a relative area that did not have blood flow per total macula area measured.

Response: We thank the reviewer for noticing this inaccuracy. This allows us to change this confusing sentence and explain what we mean by the FSV area in a much clearer way.

For each one of our images, we extract a binary mask where the different areas corresponding to the flow voids (FSV) are represented. In this sense, from this mask, we calculate two different parameters. On the one hand, we calculate the total area of FSV, corresponding this value to the sum of all relative areas that do not have blood flow in the image or, in other words, the sum of relative areas that do not have blood flow in the total measured macular area. On the other hand, we calculated the number of FSV contours corresponding to the number of flow voids regions that are present in the image.

Changes in the manuscript, methods: Following the reviewer's indication, we have rewritten and adapted the section on computational image analysis to clarify these ideas and reinforce the explanation about the extracted metrics.

3. Page 8, line 158 “Different combinations of thresholds were tested and a threshold of 30 was selected as the most appropriate. Last, a filter of the detected 160 flow voids was performed on this binary image, keeping those with an area greater than 20 pixels (0,686 μm2) after the application of a soft dilation.”

Line 163 “Different area sizes were tested when applying the size filter and the most appropriate one was selected.”

For the above parameters, how was ‘appropriate’ defined? Why was 20pixels selected as the cut-off for the FSV area?

Response: As mentioned by the reviewer, it is necessary to better specify the reason for the selection of these values, as both directly affect the obtained results.

As mentioned in the manuscript, we have tested different values for both parameters, for the thresholding and the size filter, respectively. The criterion for selecting both values is based on the optimization of the flow voids/noise ratio. The lower the thresholding value the less likely it is to detect noise as a real flow void, but this also increases the possibility of not detecting certain flow voids that are present. In the case of the filter size, the higher the parameter the higher the probability of eliminating noises but also real flow voids. Thus, we define as appropriate the values that allow detecting and segmenting flow voids with high accuracy while minimizing false positives. Based on this, we selected 30 as the threshold value, as it allows us to detect flow voids accurately with low noise. On the other hand, we select 20 pixels as a cut-off because it allows us to eliminate those flow voids that are false positives without degrading the detection of true flow voids.

Changes in the manuscript, methods: To make this idea clearer and following the reviewer's advice, we have strengthened the computational image analysis section. We now explain concisely the reasons for selecting the various parameters of our segmentation algorithm.

4. A summary flow diagram indicating the series of processes used, the two algorithms applied and decisions to establish the final method should be included. These further details for defining the method used.

Response: As indicated by the reviewer, a flowchart of the processes that are used to extract the image metrics facilitates and clarifies the functioning of our algorithm.

Changes in the manuscript: Following the reviewer's recommendation, we have added a figure (number 4) showing the effect of each step of the FSV segmentation algorithm used to perform the image analysis. In this figure, we show how the original image is transformed to obtain the binary segmentation mask from which we obtain the total FSV area and the number of contours. In addition, and in line with the points made by the reviewer in question 3, we have strengthened the computational analysis section of the image where we comment on the reasons for the different decisions that are taken to establish these steps with their different parameters. We hope that these changes will satisfy the reviewer and the usefulness of this algorithm will be better understood.

5. Page 15, Discussion. The comments on the effects of subretinal fluid in CSCR (and presumably other conditions) are very relevant and useful for clinical applications of OCTA. It would be useful to assess the same eyes with CSCR after resolution of the subretinal fluid (not just compare with unaffected other eyes of each patient). Although this is likely not practical, can the authors comment on any observations they have made post-resolution of SRF in their patients in relation to their technique and repeatability. Thank you.

Response: Thanks for the opportunity to better explain this relevant topic. We have commented on the discussion on this topic of the possible influence of the SRF on the study of the FSV, and we have included new references such as the Reich et al. study. The data from the literature support that the quantification of the FSV could be possible and reliable even with the presence of SRF, but it could slightly alter their quantification. However, the present study analyzes the reproducibility in the quantification of FSV, being high in both eyes, with and without fluid.

Regarding the analysis of the FSV after the resolution of the SRF, we did not include this in the protocol of the study because we have compared the reproducibility of the FSV between the affected eye (with SRF) and the unaffected eye (without SRF). We believe that there is also a good reproducibility of the FSV in eyes after the resolution of the SRF, which we hypothesize should be similar to the data of the unaffected eye observed in the present study.

Changes in the manuscript, discussion: Regarding CSCR, Xu et al. have described an aberrant CC flow in this disease, with a flow pattern described as focally increased and decreased pixel values, which implies a coexisting increased and decreased flow in the CC.(11) However, the main limitation of that study, as the authors recognized, was a qualitative evaluation due to the inability to get a quantitative analysis of the aberrant flow, and therefore, the lack of reproducibility of this assessment. Rochepeau et al. have demonstrated CC flow reduction in the unaffected eyes of patients with acute, recurrent, or persistent CSC at onset compared to age-matched healthy individuals, suggestive of a primary choroidopathy including ischemic processes.(8)

Yang et al. studied 56 CSCR eyes, observing a significant decrease in the total FSV area at 6 months after photodynamic therapy.(25) They have also studied an interesting issue about the possible influence of the SRF in the attenuation of FSV. They described that the relationship between the mean total area of FSV and SFCT, subfoveal CC layer thickness, and subfoveal choroidal large vessel layer thickness was not largely influenced by the presence of SRF. In the present study, we have studied two different groups, the CSCR eyes with SRF and the clinically unaffected eyes used as controls to assess the possible influence of the SRF in the FSV quantification at the CC and CH. We have observed very similar repeatability in eyes with and without SRF, so it seems that the FSV measurement is indeed a reliable parameter. Burnasheva et al. described that the presence of the general decrease of CC perfusion in both eyes of CSCR patients was irrespective of the presence of SRF or asymptomatic structural RPE changes.(30) Remarkably, as described by Byon et al. in diseased eyes, the RPE fit would generally be preferred for CC assessment, as it would essentially flatten the reference line.(16)

Therefore, as Spaide also described,(27) a reproducible quantitative analysis of the choriocapillaris was generally possible using FSV in CSCR patients with or without SRF. This is also interesting because different authors such as Teussink et al. described that the abnormal vasculature in the CC layer persisted even after the resolution of SRF.(9) In this regard, Reich et al. have described that the presence of SRF could be an important shadow-causing artifact source for CC OCTA analysis which can be mitigated but not completely eliminated by employing SS-OCTA, being the latter employed in the present study.(31)

Reich M, et al. Swept-source optical coherence tomography angiography alleviates shadowing artifacts caused by subretinal fluid. Int Ophthalmol. 2020 Aug;40(8):2007-2016. 

6. The three levels of image slabs taken all start from the RPE and are measured vertically into the underlying choriocapillaris and choroid. May have missed this, but how many scans are included for each slab? The two choriocapillaris slabs are 15 micron and 20 microns respectively, measured from the RPE/Bruch’s membrane location. The authors do mention whether taking measurements from Satller’s layer may be helpful however, based on histology, defining the outer choroidal vessel layers is not always clearly defined.

Response: The main characteristics of the 6x6 mm scan are: 500 A-scans, 500 B-scans, resolution of 12 µm, depth of 3 mm and pixel depth of 1536 pixels.

Changes in the manuscript, methods: The main characteristics of the 6x6 mm scan are: 500 A-scans, 500 B-scans, resolution of 12 µm, depth of 3 mm and pixel depth of 1536 pixels.

Changes in the manuscript, limitations: An hypothetical alternative for the analysis of the CC could be the measurements from Sattler’s layer based on histology. However, defining the outer choroidal vessel layers is not always clearly defined.

Minor comments

1. Please further clarify the exclusion criteria applied for selecting participants in the study, including details on types of pathology or disease. Were differences between right and left eyes, and gender analysed?

Response: We have provided more details on the exclusion criteria of types of pathology and disease studied. And we have included the results regarding the laterality.

Changes in the manuscript: Exclusion criteria were: ocular diseases (macular diseases such as age-related macular degeneration or macular dystrophies; retinal vascular diseases such as diabetic retinopathy, retinal vein occlusion or the presence of neovascular membrane; glaucoma and other neuropathies) or systemic pathology (such as arterial hypertension or diabetes), significant medium opacity of the lens or cornea, any ocular treatment in the three previous months or previous eye surgery.

We do not sub-analyze the results according to gender. The main reason is that CSCR affects predominantly males, as in the population studied here (60.5% males versus 39.5% females). Given that the intraclass correlation coefficient (ICC) is largely affected by the sample size, we consider that the data of males vs females should not be compared. In fact, in a pilot study, likely to the large sample size of males, they have higher ICCs values.

Changes in the manuscript, results: Regarding the laterality, 54 right and 50 left eyes were affected by CSCR, and no differences were observed in terms of the ICCs values. 

2. Line 150: “.. range of values using a mix max..” – should this be “min-max” here?

Response: We have corrected the typo. 

Changes in the manuscript, methods: First, the images were normalized to a common range of values using a min-max normalization.

3. Line 153: What are the units for the 8x8 mentioned here?

Response: We thank the reviewer for noticing this detail. We have corrected it.

The 8x8 of the contrast-limited adaptive histogram equalization refers to the number of small blocks or "tiles" into which the image is divided before calculating the histogram needed to perform the local contrast adjustment. The 8x8 would therefore be the number of local crops into which the images are divided.

Reviewer #1: Manuscript Number PONE-D-21-35561

“Repeatability of choriocapillaris flow voids by optical coherence tomography angiography in central serous chorioretinopathy"

This is a cross-sectional study which examined the intra-visit repeatability of the choroidal flow voids eyes with CSR. The study is an imaging methodological paper on choriocapillaris images in CSR eyes.

Major comments

1. Please provide the inter-grader (2 persons extracting the same slab from the same scan) and intra-grader (1 person extracting the same slab from the same scan twice) reliability.

Response: Thanks for the assessment of the manuscript performed and for the opportunity to clarify different issues.

Each of the patients included in the present study was examined by only one examiner (FJMM or JIFV). The process to acquire images is really automatic and examiner independent. The patient has to put the chin and forehead in the OCT device, after that the examiner selects the exploration mode (in this case 6x6 mm OCTA). Later, the explorer selects the button autofocus and optimize and the device center automatically the scan and focuses on the macular area. Finally, the explorer clicks on the start button for scanning the macula when the signal quality is green (>7/10). Therefore, there is no intra or inter-grader assessment of the images acquired, because is highly an automatic exploration.

The slab was predefined with an automatic specific segmentation by the software as indicated in methods (Three slabs were analyzed: inner choriocapillaris (CC) (extends 4 to 20 µm below the retinal pigment epithelium, RPE), outer CC (extends 29 to 49 µm below the RPE) and choroid (CH; measured from 64 to 115 µm below the RPE),). Correct segmentation was checked before image analysis.

No manual procedure was done to obtain the images or the appropriate segmentation, and that is why we did not assess the inter or intragrader reliability.

2. Please provide the definition or the standard of appropriate: “Different combinations of thresholds were tested and a threshold of 30 was selected as the most appropriate. Different area sizes were tested when applying the size filter and the most appropriate one was selected.”

We thank the reviewer for this comment, as we believe it is necessary to better specify why these values were selected.

We have tested different values for both parameters, both for the thresholding and the filter size, as mentioned in the manuscript. The selection criterion for the values of both is based on the optimization of the flow void/noise ratio. A lower thresholding value makes it less likely that noise will be detected as a real flow void, but this also increases the possibility of not detecting certain flow voids. For the filter size, a higher parameter increases the probability of eliminating noise but also real flow voids. Therefore, we define as appropriate the values that allow detecting and segmenting flow voids with high precision with minimized false positives. On this basis, we select 30 as the threshold value, as it allows us to detect flow voids accurately with low noise. Moreover, we select 20 pixels as a cut-off because it allows us to eliminate flow voids that are false positives without degrading the detection of true flow voids.

Following the reviewer's advice, we have reinforced the computational image analysis section. We now explain the reasons for selecting the various parameters of our segmentation algorithm in a concise manner.

3. Please provide a rationale for analysis of the three slabs at this specific layers/thickness/depth.

Response: In the discussion section, there is an explanation of the rationale to use these three specific slabs.

Changes in manuscript, discussion: It is well-known that the CC is a very thin vascular layer and the CC images currently provided by OCTA show a granular pattern of bright and dark areas of different sizes, individual capillaries in the CC not being able to be identified. (5,14) Brighter regions represent higher flow areas while dark areas are called flow voids and depict areas where there is a lack of flow signal. Besides showing comparable lateral resolution to structural OCT, OCTA can detect CC blood flow, producing a contrast between the RPE and CC.(13,14) Lin et al. have recently described that, as opposed to subfoveal choroidal thickness, there does not appear to be significant diurnal variation in CC flow voids, in terms of the density, size and numbers, in normal individuals. In their study, the CC image was taken from 31 μm to 40 μm below the RPE. Thus, the aforementioned study suggests that alterations of CC flow deficit seen in pathological eyes will not be confounded by the diurnal fluctuation.(28)

We have studied two different CC slabs to observe which one offers the most repeatable results. The first one is offered by default by the device’s software (29 to 49 µm below the RPE). More recently, a new inner slab (4 to 20 µm below the RPE) has been proposed by different authors given that the true anatomical location of the CC lies immediately under the RPE and BM.(15) In our study, we have observed higher repeatability values for the outer CC than the inner CC. This could be due to the well-known RPE signal scattering and an artifact image of the CC when the slab is segmented immediately below the RPE. The choroid slab offered good repeatability both in CSCR and fellow eyes (0.964 and 0.932) being probably less dependent on projection artifacts. However, it presents the highest CV, which means a greater level of dispersion around the mean in the CH measurements.

Byon et al. have studied 12 healthy subjects employing a 3x3-mm scan using the same device with three 10-mm-thick slabs starting 11, 21, and 31 mm below the RPE, observing an excellent ICC of 0.963, 0.975, and 0.911 respectively. They concluded that regardless of which parameter was modulated, the 21-31-µm slab was the most repeatable. Interestingly, in accordance with our study, Byon et al. have described that in some cases the most inner slab (11-21-µm) demonstrated a hypointense region caused by inadvertent inclusion of the RPE, so they decided to not include this slab in subsequent analyses.(16)

In a previous study carried out by our group using another SS-OCTA device (Triton, Topcon, Nagoya, Japan) for the reproducibility of the vessel density quantification in healthy individuals, the CC at the macula, segmented from RPE to 20.2 μm beneath it similar to the inner CC studied here, showed good repeatability in the foveal subfield (ICC=0.718).(23) However, poor repeatability was found in the parafoveal sectors (≤0.499). It should be highlighted that Yun et al. described that VD and FV areas of the CC varied according to the device used and the image adjustment method.(29)

4. Please define FSV area – is this the size of single flow void or the total area of flow void in the scan. The authors have only showed the FSV area. Please also show the density of flow void.

Response: We are thankful to the reviewer for this. It is clear that the definition of the FSV area is unclear and should be clarified.

For each one of our images, we extract a binary mask where the different areas corresponding to the flow voids (FSV) are represented. In this regard, we calculate two different parameters from this mask. First, we calculate the total area of FVS, this value corresponds to the sum of all the relative areas that do not have blood flow in the image or, in other words, the sum of relative areas that do not have blood flow in the total macular area measured. In addition, we calculated the number of FSV contours that corresponds to the number of flow voids regions present in the image.

Thus, as recommended by the reviewer, we have adapted and revised the section on computational image analysis to clarify these issues and to reinforce the explanation of the extracted metrics.

5. Please compare the results between normal and CSR eyes.

Response: Thanks for the suggestion.

Changes in the manuscript, results section: 

Similar ICC values were observed between CSCR eyes and the fellow eyes (0.859, 0.959 and 0.964 versus 0.844, 0.949 and 0.932 for the inner CC, outer CC and choroid respectively).

In the comparison between CSCR eyes and fellow eyes for the inner and outer CC measurements, similar ICCs values were observed (0.832 and 0.884 versus 0.843 and 0.907 respectively). However, a noticeably higher ICC value was observed for the choroid measurements of diseased eyes than for the unaffected ones (0.935 vs 0.881 respectively).

Changes in the manuscript, discussion: Similar ICC values were observed between CSCR eyes and unaffected fellow eyes, except for the choroid in which higher values were observed for the disease’s eyes.

6. Please also show the Bland–Altman plots of the different choriocapillaris flow void metrics.

Response: As suggested, we have included two new figures showing the Bland-Altman plots of the different flow void metrics and segmentations.

Fig. 5: Bland-Altman plots showing the flow signal voids (FSV) area in the different segmentations studied (A and D: inner choriocapillaris (CC) in CSCR and fellow eyes respectively; B and E: outer CC in CSCR and fellow eyes respectively, C and F: choroid in CSCR and fellow eyes respectively).

Figure 6: Bland-Altman plots showing the flow signal voids (FSV) number of contours in the different segmentations studied (A and D: inner choriocapillaris (CC) in CSCR and fellow eyes respectively; B and E: outer CC in CSCR and fellow eyes respectively, C and F: choroid in CSCR and fellow eyes respectively).

Minor comments

7. Please provide a step-by-step flow chart on how the image processing was carried out.

Following the reviewer's recommendation, we have added a figure showing step-by-step how the image processing is carried out (figure 4).

8. Were the two scans aligned before the analysis? Previous study (PMID: 31833241) used a customized software to align the scan prior to analysis. Was the eye tracker turned on during the scanning of the two consecutive macular scans?

Response: The eye tracker was activated so the exploration was automatically centered on the same area. Therefore, the two scans were aligned before the analysis.

Changes in the manuscript, methods: The eye tracker was activated so the exploration was automatically centered on the same area. Therefore, the two scans were aligned before the analysis.

9. Please elaborate the types of pathologies excluded from the study. “Exclusion criteria were ocular or systemic pathology…”

Response: We have provided more details on the exclusion criteria of types of pathology and disease studied.

Changes in the manuscript: Exclusion criteria were: ocular diseases (macular diseases such as age-related macular degeneration or macular dystrophies; retinal vascular diseases such as diabetic retinopathy, retinal vein occlusion or the presence of neovascular membrane; glaucoma and other neuropathies) or systemic pathology (such as arterial hypertension or diabetes), significant medium opacity of the lens or cornea, any ocular treatment in the three previous months or previous eye surgery.

10. Were scans with other types of artifacts i.e., motion or poor segmentation excluded from the study? “Only images of sufficient quality, as determined by a signal quality >7/10, were accepted.”

Response: Thanks for the comment. Two cases were excluded because of a lack of collaboration which produces motion artifacts.

Changes in the manuscript, methods: Exclusion criteria were: ocular diseases (macular diseases such as age-related macular degeneration or macular dystrophies; retinal vascular diseases such as diabetic retinopathy, retinal vein occlusion or the presence of neovascular membrane; glaucoma and other neuropathies) or systemic pathology (such as arterial hypertension or diabetes), significant medium opacity of the lens or cornea, any ocular treatment in the three previous months or previous eye surgery. In addition, images with large motion artifacts due to the lack of collaboration and the impossibility to activate the eye tracker by the software were excluded.

Changes in the manuscript, results: Two cases were excluded because of a lack of collaboration which resulted in motion artifacts, being the final sample size analyzed one hundred and four eyes of 52 patients with unilateral CSCR, including 52 affected eyes (with the presence of SRF) and 52 eyes clinically unaffected.

11. Figure 2. Please clarify exploration 1 and 2.

Response: As requested, we have clarified this issue in the legend of figure 2. Exploration refers to the two consecutive scans separated by a two-minute interval.

Fig. 2: Optical coherence tomography angiography (OCTA) images and the corresponding analyzed images of the flow signal voids (FSV) area of the three slabs studied (inner choriocapillaris, CC; outer CC and choroid) for the exploration 1 and 2 (two consecutive scans separated by a two-minute interval) in central serous chorioretinopathy (CSCR) and fellow eyes.

---

## [Decision Letter · Decision Letter 1]

5 Dec 2022

Repeatability of choriocapillaris flow voids by optical coherence tomography angiography in central serous chorioretinopathy.

PONE-D-21-35561R1

Dear Dr. Fernández-Vigo,

We’re pleased to inform you that your manuscript has been judged scientifically suitable for publication and will be formally accepted for publication once it meets all outstanding technical requirements.

Kind regards,

Michele Madigan

Academic Editor

PLOS ONE

Additional Editor Comments (optional):

Reviewers' comments:

Reviewer's Responses to Questions

**Comments to the Author**

1. If the authors have adequately addressed your comments raised in a previous round of review and you feel that this manuscript is now acceptable for publication, you may indicate that here to bypass the “Comments to the Author” section, enter your conflict of interest statement in the “Confidential to Editor” section, and submit your "Accept" recommendation.

Reviewer #1: All comments have been addressed

2. Is the manuscript technically sound, and do the data support the conclusions?

Reviewer #1: Yes

3. Has the statistical analysis been performed appropriately and rigorously? 

Reviewer #1: Yes

4. Have the authors made all data underlying the findings in their manuscript fully available?

Reviewer #1: No

5. Is the manuscript presented in an intelligible fashion and written in standard English?

Reviewer #1: Yes

6. Review Comments to the Author

Reviewer #1: All the comments have been well addressed. No further comments. Appreciate the authors' detailed revision.

7. PLOS authors have the option to publish the peer review history of their article (what does this mean?). If published, this will include your full peer review and any attached files.

Reviewer #1: No

---

## [Editor Report · Acceptance letter]

8 Dec 2022

PONE-D-21-35561R1 

Repeatability of choriocapillaris flow voids by optical coherence tomography angiography in central serous chorioretinopathy. 

Dear Dr. Fernández-Vigo:

I'm pleased to inform you that your manuscript has been deemed suitable for publication in PLOS ONE. Congratulations! Your manuscript is now with our production department. 

Kind regards, 

on behalf of

Dr. Michele Madigan 

Academic Editor

PLOS ONE